Pine polyphenols from Pinus koraiensis prevent injuries induced by gamma radiation in mice

Li Hui 1
Wang Zhenyu 1 wangzy219001@yahoo.com.cn
Xu Yier 2
Sun Guicai 3
1 School of Chemistry and Chemical Engineering, Harbin Institute of Technology , Harbin, Heilongjiang , China
2 Department of Pharmacology, Pharmaceutical Academy of Harbin Pharmaceutical Group , Harbin, Heilongjiang , China
3 Department of Orthopaedics, The Fourth Hospital Affiliated to Nanchang University , Nanchang, Jiangxi , China
Ng Tzi Bun
Electronic publication date: 2016 Apr 5
Publication date: 2016
Volume: 4
Electronic Location ID: e1870
Received 2016 Jan 2; Accepted 2016 Mar 9
Copyright: ©2016 Li et al.
Copyright year: 2016
Copyright holder: Li et al.
License: This is an open access article distributed under the terms of the Creative Commons Attribution License, which permits unrestricted use, distribution, reproduction and adaptation in any medium and for any purpose provided that it is properly attributed. For attribution, the original author(s), title, publication source (PeerJ) and either DOI or URL of the article must be cited.
License URL: https://creativecommons.org/licenses/by/4.0/

Keywords: Pine polyphenols, γ-radiation, Antioxidant, Redox, Apoptosis

Funding: National Natural Science Foundation of China 31170510 81160445 Financial support was provided by the National Natural Science Foundation of China (NO. 31170510) and the National Natural Science Foundation of China (NO. 81160445). The funders had no role in study design, data collection and analysis, decision to publish, or preparation of the manuscript.

==============================
Pine polyphenols (PPs) are bioactive dietary constituents that enhance health and help prevent diseases through antioxidants. Antioxidants reduce the level of oxidative damages caused by ionizing radiation (IR). The main purpose of this paper is to study the protective effect of PPs on peripheral blood, liver and spleen injuries in mice induced by IR. ICR (Institute of Cancer Research) male mice were administered orally with PPs (200 mg/kg b.wt.) once daily for 14 consecutive days prior to 7 Gy γ-radiations. PPs showed strong antioxidant activities. PPs significantly increased white blood cells, red blood cells and platelets counts. PPs also significantly reduced lipid peroxidation and increased the activities of superoxide dismutase, catalase and glutathione peroxidases, and the level of glutathione. PPs reduced the spleen morphologic injury. In addition, PPs inhibited mitochondria-dependent apoptosis pathways in splenocytes induced by IR. These results indicate that PPs are radioprotective promising reagents.

Introduction

Polyphenols, divided into phenolic acids, flavonoids and tannins, are bioactive dietary constituents widely spread throughout the plant kingdom and most abundant in coffee, tea, red wine, some vegetables and fruits. Human consumption studies indicate that 500–1,500 mg of polyphenols are frequently consumed per day, and it is not anticipated that any acute or lethal toxicity would be observed by the oral intake route (Vogiatzoglou et al., 2014). Polyphenols can enhance health and help prevent chronic diseases (Birt & Jeffery, 2013), such as hypoglycemic (Son et al., 2013), hypolipidemic (Bao et al., 2014) and visceral fat reduction (Heber et al., 2014).

Ionizing radiation (IR) is a common treatment modality against various cancers; however, it also damages normal cells and tissues (Stone et al., 2003). The risk of IR is rapidly increasing. Exposure to IR damages biological macromolecules such as proteins, lipids and DNA in direct and indirect pathways. IR also triggers the radiolysis of water in the cellular system and induces the generation of extraordinary high levels of reactive oxygen species (ROS) in milliseconds, which induces immediate and widespread oxidative damages (Sagar, 2005). ROS causes a redox imbalance in cells and living tissues. Although endogenous cellular antioxidants such as superoxide dismutase (SOD), glutathione peroxidase (GSH-Px), catalase (CAT) and reduced glutathione (GSH) act in concert to eliminate ROS accumulation in a physiological state, under pathological conditions ROS overload might exceed the cellular antioxidant capacity, affecting critical biological macromolecules and triggering oxidative stress (Guelman et al., 2005).

Antioxidants reduce the level of oxidative damages caused by free radicals and ionizing radiation (Oršolić & Bašić, 2005). Pine polyphenols is distributed in pine skin, pine needles and pine cones; they are antioxidant (Ugartondo et al., 2007), anti-tumor (Hsu et al., 2005), anti-inflammatory (Ince et al., 2009), antidiabetic (Zibadi et al., 2008), anti-UV radiation (Kyriazi et al., 2006), etc. The main active ingredient of pine polyphenols is catechin-based flavonoids and phenolic acids. Polyphenols from Pinus koraiensis have a strong antioxidant activity in ABTS⋅+ radical (Li & Wang, 2015). Fractionation of pine bark extracts in Pinus pinaster and P. radiata contained mainly flavan-3-ol monomers and procyanidins oligomers and have antioxidant activity in DPPH stable radicals (Jerez, Sineiro & Nuñez, 2009). Pine bark extracts, obtained by water extraction of the raw bark of the P. maritime, contain mainly catechin, epicatechin, taxifolin and procyanidins and have free radical scavenging activity such as the stable radical DPPH and the oxygen free radicals O2⋅− and HO⋅, and act as a protective factor against ultraviolet (UV)-radiation-induced injury by virtue of its antioxidant capacity (Packer, Rimach & Virgili, 1999). IR is a routine treatment modality against various cancers; it also damages normal cells and tissues around the tumor. Polyphenol has the potential for new drugs in antitumor and anti-radiation treatment at the same time; it can effectively reduce normal tissue damage caused by radiotherapy. In recent years, interest in exploring the radioprotective potentials of plants and phytochemicals has escalated (Arora, 2008). For our studies, we have evaluated the radioprotective property of varieties of plants with polyphenols, flavonoids and anthocyanin. However, it is rarely reported that pine polyphenols (PPs) from Pinus koraiensis have a radioprotective effect in mice. This article is the first attempt to investigate the radioprotective efficacy of PPs against γ-radiation-induced peripheral blood, liver and spleen injuries in mice.

Materials and Methods

Chemicals

Phenazine methosulphate (PMS), nitrotetrazolium blue chloride (NBT) and 2, 2-diphenyl-1-picrylhydrazyl (DPPH) were purchased from Sigma (St. Louis, MO, USA). Nicotinamide adenine dinucleotide (NADH) was purchased from Roche (Basel, Switzerland). D101 macroporous resin was obtained from the Chemical Plant of Nankai University (China). Food grade 95% ethanol was purchased from a local reagent corporation. The total protein (BCA, NO: A045-3), SOD (NO: A001-1), GSH-Px (NO: A005), CAT (NO: A007-1), MDA (NO: A003-1) and GSH (NO: A006-2) assay kits were purchased from Nanjing Jiancheng (China). Rabbit anti-Bcl-2 (sc-492, 1:200), Bax (sc-526, 1:200), cytochrome c (sc-7159, 1:200) and caspase-3 (sc-7148, 1:200) polyclonal antibodies were purchased from Santa Cruz (Santa Cruz, CA, USA), Rabbit anti-β-actin (AC006, 1:2,000) polyclonal antibody were purchased from ABclonal (USA) and HRP-conjugated goat anti-rabbit IgG (ZB-2301,1:20,000) polyclonal antibody were purchased from ZSGB-BIO (Beijing, China). Potent ECL was purchased from Multisciences (Hangzhou, China). All other chemicals with the highest purity grade available were purchased from reputed local manufacturers/suppliers.

PPs was perpetrated according to our previous method (Li & Wang, 2015) and was again enriched with the chromatographic column of D101 macroporous resins. Identified by HPLC, its main ingredient is catechin-3-O-glucose, which accounted for more than half. Other ingredients include catechin, epicatechin, massonianoside B, catechin 3-O-rutinoside, cedrusin and massonianoside C.

Antioxidant assay

DPPH free radical scavenging activity

DPPH free radical scavenging activity was measured by means of the absorbance of DPPH at 517 nm (Xia et al., 2014). Briefly, 1.0 mL sample solution was added to 1.0 mL of DPPH solution (0.2 mM in methanol). The decrease in the solution absorbance at 517 nm was measured after 30 min of incubation. DPPH free radical scavenging activity was calculated using the following formula: DPPH free radical scavenging activity%=1−A1∕A0×100

where A0 is the absorbance of DPPH free radicals without sample, and A1 is the absorbance of DPPH free radicals with samples. The efficient concentration of samples that scavenges 50% of DPPH free radical (EC50) was calculated and expressed as µg/mL.

Superoxide anion radical scavenging activity

The superoxide anion radical scavenging activity was established by monitoring the competition of those with NBT for the superoxide anion generated by the PMS–NADH system (Biswas, Chatli & Sahoo, 2012). The reduction mixture contained 150 µL NBT (100 µM), 450 µL NADH (100 µM) and a sample solution 200 µL. Total volume was made up to 1 mL with distilled water and then 1.9 mL of Tris–HCl buffer (0.02 M, pH 8.0) was added. The reaction was started by adding 100 µL of PMS (100 µM) and then the change in absorbance (A) was recorded at 560 nm after 1 min at 37 °C. The superoxide anion free radical scavenging activity was calculated with the following equation: Superoxide anion radical scavenging activity%=1−A1∕A0×100

where A0 is the absorbance of superoxide anion radicals without sample; and A1 is the absorbance of superoxide anion radicals with sample. The efficient concentration of samples that scavenges 50% of the superoxide anion radicals (EC50) was calculated and expressed as µg/mL.

Hydroxyl radical scavenging activity

The hydroxyl radical scavenging activity was investigated using Fenton’s reaction (Fe2+ + H2O2 → Fe3+ + OH− + OH⋅) (Jiang, Chen & Shi, 2013). The reaction mixture containing 1 mL sample solution was incubated at 37 °C for 2 h with 1 mL of 9 mM salicylic acids, 1 mL of 9 mM FeSO4, and 1 mL of 8.8 mM H2O2, and then the absorbance was read at 510 nm. The hydroxyl radical scavenging activity was calculated from the following equation: Hydroxyl radical scavenging activity%=1−Ai−Aj∕A0×100

where A0 is the absorbance of the hydroxyl radical with a treated control. Ai and Aj are the absorbances of the hydroxyl radical with the treated sample, and the absorbance of the hydroxyl radical with the non-treated sample. The efficient concentration of samples that scavenges 50% of the hydroxyl radicals (EC50) was calculated and expressed as µg/mL.

Reducing power (RP) assay

The RP assay was conducted (Seo et al., 2012). The reducing power of the antioxidant is measured by the transformation of the Fe3+∕ferricyanide complex into the ferrous form. A freshly prepared sodium phosphate buffer (1.0 mL, 0.2 M, pH 6.6) and 1% K3Fe(CN)6 (1.0 mL) were added to the sample solution (1.0 mL). After incubating the mixtures at 50 °C for 20 min, 10% trichloroacetic acid (1.0 mL) was added. The resulting mixtures were centrifuged at 3,000 rpm for 10 min at 4 °C . The upper layer (1.0 mL) was diluted with water (1.0 mL), and 0.1% FeCl3 (0.5 mL) was hereby added. The absorbance of the resultant solution was measured at 700 nm. The efficient concentration of samples that were of 0.5 absorbance value (RP0.5) was calculated and expressed as µg/mL.

Animals

Male ICR (Institute of Cancer Research) mice, 4-6-week-old with body weight 20 ± 2 g were obtained from the Harbin Medical University (Harbin, China). The mice were housed in a mouse room at room temperature with a 12-h light/dark cycle and were provided with free access to standard mouse chow and water ad libitum. The experimental protocols were approved by Heilongjiang University of Chinese Medicine (SCXK Hei 2008004). All efforts were made to minimize animal suffering.

Screening of the dose of PPs

Mice were divided into five groups of six animals each. PPs (50, 100 and 200 mg/kg bwt/d) were given by oral administration for 14 consecutive days prior to irradiation. The control group and the IR group were given with the equal volume of normal saline every day. Irradiation was performed at a dose rate of 1.3 Gy/min using a 60Co irradiator (Heilongjiang Academy of Agricultural Sciences, China). The irradiation dose was 7 Gy (Park et al., 2008). The source-to-mice distance was 400 cm at room temperature. After 24 h, the mice were sacrificed by cervical dislocation and their spleen were excised and weighted.

Spleen index was calculated according to the following equation: Spleen index=Spleen weight/body weight×100.

Groups and irradiation

Mice were divided into four groups of ten animals each, detailed below:

Control group—normal saline + Sham irradiation.

PPs group—200 mg/kg bwt/d PPs + Sham irradiation.

IR group—normal saline + 7 Gy 60Co γ-irradiation.

PPs + IR group—200 mg/kg bwt/d PPs + 7 Gy 60Co γ-irradiation.

PPs were given by oral administration for 14 consecutive days prior to irradiation. The irradiation dose was 7 Gy. After 24 h, the mice were sacrificed by cervical dislocation and their eyeball blood, liver and spleen were excised and weighted. The blood was centrifuged for 15 min at 3,500 rpm in 4 °C . Supernatant was plasma. 10% (w∕v) homogenates were prepared in normal saline.

Hematological parameters

After 24 h post-irradiation, the eyeball blood of mice was collected and analyzed by animal automatic blood analyzer. White blood cells (WBC), red blood cells (RBC) and platelets were counted.

Spleen lymphocyte transformation assay

Spleens were removed after 24 h post-irradiation and spleenocytes were isolated according to a previously described procedure. Briefly, spleens were cut into small pieces and then expelled through a 200 mesh nylon cell strainer. After washing the cell strainer with 10 mL RMIP-1640, the cells were pelleted at 800× g for 5 min. The cell pellet was resuspended in 1 mL ACK lysis buffer and incubated for 5 min to lyse erythrocytes. Subsequently, lysis was stopped by adding 9 mL of RMIP-1640; the cells were pelleted at 800× g for 5 min, resuspended in RMIP-1640 with 10% FBS, and incubated for 30 min at 37 °C in a regular CO2-controlled incubator to allow cells to recover from isolation. After checking spleenocyte viability by a trypan blue dye exclusion test, 1 × 106 cells/mL were seeded in 96-well plates, half of which added lipopolysaccharide (LPS, 10 µg/mL) as mitogen, and incubated for 72 h at 37 °C in a humidified chamber with 5% CO2. Cell vitality was determined by MTT. Spleen cell transformation=ODLPS−ODControl∕ODControl×100.

Antioxidant status assessment

The SOD, CAT, GSH-Px activities and the levels MDA and GSH in plasma, 10% liver and spleen homogenates were determined according to experimental procedure provided by manufacturers. Tissue protein was estimated according to the kit (BCA method) using BSA as standard.

Histopathology assay of spleen tissues

Spleen tissues were fixed in 4% paraformaldehyde solution, and then processed by routine techniques for embedding in paraffin. Blocks were sectioned at a thickness of 5 µm and stained with hematoxylin and eosin for a histopathological examination, which were performed under a light microscopy and documented by an Olympus microphotocamera.

Western blotting

The expression levels of Bcl-2, Bax, caspase-3 and cytochrome c proteins in spleen were examined by Western blotting. 10% of spleen homogenates were prepared by T10 basic homogenizer (IKA, Staufen, Germany) in RIPA Buffer containing 1 mM PMSF. Denatured protein samples (20 µg) were subjected to electrophoresis in a 5–10% SDS-PAGE and then transferred onto a PVDF membrane. After treated with PBS-T Blocking Buffer, PVDF membranes were incubated overnight at 4 °C with primary antibodies for Bcl-2, Bax, caspase-3, cytochrome c or β-actin. After incubated with HRP linked secondary antibodies in PBS-T, chemiluminescent reaction was developed by an ECL hydrogen peroxide solution and detected in the Image Lab System (Bio-Rad, Hercules, CA, USA).

Statistical analysis

Multiple group comparison was done using one-way analysis of variance followed by the Tukey post hoc test using SPSS10. All values were represented as mean ± S.D. Values of P < 0.05 were considered significant.

Results

PPs show the strong antioxidant activities in vitro

DPPH radical, superoxide anion radical, hydroxyl radical scavenging activities and reducing power assay of PPs were assayed. PPs showed strong antioxidant activities in DPPH radical (EC50 = 324.0 ± 13.52μg/mL), superoxide anion radical (EC50 = 117.1 ± 6.13μg/mL), hydroxyl radical (EC50 = 519.9 ± 25.91μg/mL) and reducing power assay (RP0.5 = 363.2 ± 5.34μg/mL) (Fig. 1 and Table S1). DPPH radical, superoxide anion radical scavenging activities and reducing power assay of PPs were lower than catechin and gallic acid; hydroxyl radical scavenging activity was lower than gallic acid, but higher than catechin. Here, PPs show strong antioxidant activities in vitro.

Figure 1 Antioxidant activities of PPs from Pinus koraiensis.

(A) DPPH free radical scavenging activity; (B) superoxide anion radical scavenging activity; (C) hydroxyl radical scavenging activity; (D) reducing power assay. Each value represents mean ± SD (repeat three times).

The optimal dose of PPs

Radioprotective effects of PPs with different doses were investigated. Radiation dose 7 Gy was supposed to radiate the mice (Park et al., 2008). The spleen is radiation sensitive organ, and the spleen index analysis dose–response effect. The results were presented in Fig. 2. PPs of different doses showed radioprotective effect. 200 mg/kg bwt/d PPs showed the best radioprotective effect.

PPs protect the hematological parameters in irradiated mice

IR significantly decreased WBC, RBC and platelets counts in mice in comparison to the control group (Table 1), from 3.53 ± 0.323 × 109∕L, 9.88 ± 0.866 × 1012∕L and 511.6 ± 40.41 × 109∕L to 0.61 ± 0.052 × 109 (P < 0.05), 6.40 ± 0.463 × 1012 (P < 0.05) and 61.7 ± 5.92 × 109 (P < 0.05) respectively; PPs reduced the loss of WBC, RBC and platelets counts compared to the IR group. PPs alone had no significant influence on WBC, RBC and platelets counts. But platelets counts in PPs group were lower than the control group (not significant). Therefore, PPs have an antagonistic effect on radiation-induced peripheral blood damage.

Table 1 WBC, RBC and platelets counts, spleen index and spleen lymphocyte transformation in mice.

Groups	WBC (109/L)	RBC (1012/L)	Platelets (109/L)	Spleen index	Transformation (%)	
Control	3.53 ± 0.323	9.88 ± 0.866	511.6 ± 40.41	2.89 ± 0.352	30.59 ± 4.112	
PPs	3.64 ± 0.244	9.91 ± 0.927	471.7 ± 31.53	3.02 ± 0.374	27.52 ± 3.523	
IR	0.61 ± 0.052*	6.40 ± 0.463*	61.7 ± 5.92*	1.31 ± 0.131*	8.33 ± 1.123*	
PPs + IR	1.03 ± 0.083*	9.10 ± 0.832**	370.9 ± 33.86*,**	1.92 ± 0.283*	23.89 ± 2.031**	
Notes.

PPs of 200 mg/kg bwt/d were given by oral administration for 14 consecutive days prior to irradiation (7 Gy). Each group was of ten animals except spleen lymphocyte transformation (n = 4), and every group experiments were duplicate.

* P < 0.05 vs control.

** P < 0.05 vs IR.

n is number of mice.

PPs show the immunomodulatory effect in irradiated mice

The immunomodulatory effect in irradiated mice was assayed by spleen index and spleen lymphocyte transformation. Irradiation significantly decreased spleen index and spleen lymphocyte transformation from 2.89 ± 0.352 and 30.59 ± 4.112 to 1.31 ± 0.134 (P < 0.05) and 8.33 ± 1.122 (P < 0.05) respectively; PPs increased the spleen index and spleen lymphocyte transformation compared to the IR group (Table 1). Here, PPs strengthened the spleen lymphocyte transformation proliferation. Thus, PPs show immunomodulatory effects in irradiated mice.

PPs inhibit the lipid peroxidation in irradiated mice

Malondialdehydes (MDA) are biomarkers of lipid peroxidation. Irradiation significantly increased the level of MDA in plasma, liver and spleen; PPs pretreatment significantly reduced the level of radiation-induced MDA (Fig. 3). Thus, lipid peroxidation increases after irradiation whereas PPs ameliorate the change.

Figure 2 The radioprotective effect of different doses PPs pretreatment on spleen index (n = 6) in mice. The irradiation dose was 7 Gy. ∗P < 0.05 vs Control, #P < 0.05 vs IR. n is number of mice.

Figure 3 The effect of PPs (200 mg/kg bwt/d) pretreatment on the levels of MDA on plasma (n = 10), liver (n = 10) and spleen (n = 6) in mice. The irradiation dose was 7 Gy. ∗P < 0.05 vs Control, #P < 0.05 vs IR. n is number of mice.

Figure 4 The effect of PPs (200 mg/kg bwt/d) pretreatment on the levels of enzymatic and non-enzymatic antioxidants on plasma (n = 10), liver (n = 10) and spleen (n = 6) in mice. The irradiation dose was 7 Gy. (A) SOD; (B) CAT; (C) GSH-Px; (D) GSH. ∗P < 0.05 vs Control, #P < 0.05 vs IR. n is number of mice.

PPs increase the SOD, CAT, GSH-Px activities and the level of GSH in irradiated mice blood, liver and spleen

SODs comprise a family of metal-containing proteins that catalyze dismutation of superoxide anion (O2−) to form H2O2 and O2. CATs are heme-containing enzymes that convert H2O2 into H2O and O2. GSH-Px inactivates peroxides by using GSH as a source of reducing equivalents. GSH is the most abundant intracellular antioxidant which prevents protein thiol groups from oxidation, either by directly reacting with reactive species or indirectly through GSH-Px. Irradiation significantly decreased SOD, CAT, GSH-Px activities and the level of GSH in plasma, liver and spleen; PPs pretreatment interdictted radiation-induced loss of SOD, CAT, GSH-Px activities and the level of GSH (Fig. 4). Thus, PPs prevent radiation-induced redox imbalance by protecting the SOD, CAT, GSH-Px activities and the level of GSH.

PPs weaken the spleen injury in morphology in irradiated mice

The splenic morphology was characterized by hematoxylin and eosin staining. The results were shown in Fig. 5. The spleen is made of red pulp and white pulp. Red pulp contains T lymphocytes, and white pulp contains B lymphocytes. Lymphocytes belong to the radiation sensitive cells. Radiation exposure significantly reduced the cellularity of white pulp, decreased the width and the density of the layer of lymphocytes, and reduced the cellularity of red pulp and accompanied by tissue congestion. Compared with IR group, PPs pretreatment before IR caused lower damage in the cellularity of white pulp and red pulp. Thus, PPs prevent radiation-induced spleen morphologic injury by protecting splenic lymphocytes.

Figure 5 Photomicrographs of spleen sections stained with hematoxylin and eosin staining. PPs were given at a dose of 200 mg/kg body weight daily for 14 consecutive days prior to irradiation (7 Gy). Spleen histology: magnification: 10×. (A) Spleens of mice with the Control are comprised of both red (RP) and white pulps (WP). (B) Spleens in mice with PPs. (C) Spleens in mice with IR. (D) Spleens of mice in mice with PPs + IR.

PPs regulate the expression level of apoptosis-related proteins in irradiated mice spleens

A western blot analysis of specific apoptosis-related proteins was carried out to determine whether PPs had any effect on their expression levels. Irradiation increased the expression of Bax, cytochrome c and caspase-3 proteins accompanying a decrease in the expression of Bcl-2 protein. PPs restrained irradiation induced expression increase of Bax (Figs. 6A and 6C), cytochrome c (Figs. 6A and 6E) and caspase-3 (Figs. 6A and 6F) proteins substantially and promoted the expression of Bcl-2 (Figs. 6A and 6B) protein, thus decreased the ratio of Bax/Bcl-2 (Figs. 6A and 6D), from 4.76 to 2.70 times higher than those of the control group. PPs alone treatment did not significantly affect expression level of apoptosis-related proteins. These findings suggest that PPs presumably prevent radiation-induced mitochondria-dependent apoptosis pathways.

Figure 6 PPs (200 mg/kg bwt/d) inhibits the expression levels of Bax, cytochrome c and caspase-3 induced by irradiation (7 Gy) and inceases the expression levels of Bcl-2 in spleen (n = 6). (A) The expression of Bcl-2, Bax, cytochrome c and caspase-3 in spleen by Western blotting. (B) The expression level of Bcl-2. (C) The expression level of Bax. (D) The ratio of Bax/Bcl-2. (E) The expression level of cytochrome c. (F) The expression level of caspase-3. ∗P < 0.05 vs Control, #P < 0.05 vs IR. n is number of mice.

Discussion

The main aim to find a suitable radio-protectant from phytochemicals is that it should ameliorate the mice peripheral blood, liver and spleen injuries after radiation exposure. The liver is the main metabolism and detoxification organ, the spleen is a center of activity of the mononuclear phagocyte system and blood can effectively reflect the body’s metabolism and health. Therefore the peripheral blood, the liver and the spleen were chosen to evaluate the efficacy of the radiation protection of PPs.

The antioxidant analysis found that PPs have strong antioxidant activities and efficiently scavenged the free radicals such as superoxide anion, hydroxyl radical and DPPH radical in a dose-dependent manner. Superoxide anion and hydroxyl radical belongs to the active oxygen free radicals, which can be formed by IR. Flavonoids, including epicatechin, rutin, catechin, epigallocatechin, quercetin, etc, can scavenge superoxide anions and hydroxyl radical (Hort et al., 2008). Pine (Pinus pinaster and Pinus radiate) bark fractions showed antioxidant activity (Touriño et al., 2005; Jerez et al., 2011). The water soluble extracts of Korean black pine (Pinus thunbergiana), catechin, epicatechin, quercetin and ferulic acid as active compounds, showed the radical scavenging activities of DPPH and reductive potential of ferric ion (Shen et al., 2010). A wide range of antioxidant phytochemicals, including flavonoids, and polyphenols, are antioxidants that are radioprotective in experimental systems.

The RBCs carry oxygen from the respiratory organs to the rest of the body (Epstein & Hsia, 1998). WBCs are the cells of the immune system and are produced and derived from bone marrow a hematopoietic stem cell. Platelets are fragments of cytoplasm which are derived from the megakaryocytes of the bone marrow (Machlus, Thon & Italiano, 2014). IR causes bone marrow suppression, which suppress the hematopoietic stem and progenitor cells proliferation and differentiation (Wang et al., 2006; Meng et al., 2003). The rapidly dividing cells of the blood system, especially leukocytes and erythrocytes, and the immune organs and immune cells are highly sensitive to IR (Widel et al., 2003; Yang et al., 2012). Radiation significantly decreased peripheral blood WBC, platelet and RBC counts (Jiang et al., 2015; Suryavanshi et al., 2015). The PPs pretreatment significantly decreased radiation-induced damage in RBC, WBC and platelet counts. Previously reports have shown that tea polyphenols rich in catechins were given oral administration at 50 and 100 mg/kg body weight and had a radioprotective effect on the hematopoietic system of radiation-induced damage in mice (Hu et al., 2011). RBCs, WBCs, and platelets all come from bone marrow hematopoietic system, which showed that PPs may protect bone marrow hematopoietic system against radiation-induced damage.

The immune system is one of the most important defense mechanisms against IR (Zhao et al., 2014). The damage degree of normal tissue in radiation is dependent on the dose, tissue sensitivity, repair capacity, and prevailing endogenous antioxidant defenses. Spleen tissue is one of the most seriously damaged organs by radiation, which may be because spleen tissue contains a high proportion of lymphocytes. The spleen synthesizes antibodies in its white pulp and contains half of the body’s monocytes within the red pulp (Swirski et al., 2009). Radiation severely damaged the spleen lymphocytes and reduced the size and weight of spleen (Koo et al., 2013). Eckol significantly decreased the mortality of lethally irradiated mice by an improvement in hematopoietic recovery, the repair of damaged DNA in immune cells and an enhancement of their proliferation (Park et al., 2008). The PPs pretreatment significantly decreased radiation-induced damage in spleen index and spleen lymphocytes transformation. The IR group revealed an enlarged fused white pulp with increased sinusoidal spaces; the matrix was completely destroyed when compared to the control group. Pretreatment with PPs prevented the radiation-induced spleen damage. So PPs, like as other polyphenols, prevented radiation-induced injury by immunomodulatory effects.

IR passing through living tissues generates excessive ROS, which causes lipid peroxidation, damages redox homeostasis within cells and living tissues, generates MDA, and decreases the levels of enzymatic and non-enzymatic antioxidants (Sagar, 2005; Rodeiro, Delgado & Garrido, 2014; Kumar et al., 2015). IR inhibited the expression of SOD1, SOD2, CAT, and GPX1 mRNA in hematopoietic stem cells and the enzyme activity of SOD, CAT, and GPX1 in bone marrow cells (Xu et al., 2015). Under normal physiological conditions, the levels of enzymatic (SOD, CAT, GSH-Px) and non-enzymatic (GSH) antioxidants could prevent or limit oxidative damages. Here, the pretreatment with PPs reduced the levels of MDA and restored the SOD, CAT, GSH-Px activities and the level of GSH induced by radiation. These restored SOD, CAT, GSH-Px and GSH additionally play an essential role to fight ROS induced by IR. The decrease in the activities of SOD, CAT and GSH-Px in irradiated mice is due to the inhibition or oxidative inactivation of enzyme caused by ROS generation, which in turn can impair the antioxidant defense mechanism (Jagetia & Reddy, 2005). The depletion of GSH in mice is known to inhibit GSH-Px activity and has been shown to increase lipid peroxidation (Nair, Parida & Nomura, 2001). Acorus calamus extract (250 mg/kg body weight), rich in polyphenols, significantly increased the enzyme activity of the antioxidant defense system, especially SOD, CAT and GPx and the level of GSH and decreased the formation of MDA (Sandeep & Nair, 2012). PPs reduced the radiation-induced redox imbalance and lipid peroxidation in plasma, spleen and liver tissue, and partly restored redox balance. Many studies have found that the protection of redox balance in cells and tissues is one of the important mechanisms of polyphenol in anti-radiation and anti-oxidative stress (Yagi, Tan & Tuan, 2013; Ghosh, Dey & Saha, 2014; Srinivasan et al., 2006).

Apoptosis represents a universal and exquisitely efficient suicide pathway. Radiation caused apoptosis in blood lymphocytes and splenocytes (Zhu et al., 2014). Here, IR induced in a mitochondria-dependent cell death in splenocytes (Park et al., 2011). Furthermore, PPs pretreatment before IR to mice decreased in the expression levels of Bax and increased in the expression levels of Bcl-2, thus reducing the Bax/Bcl-2 ratio, which favors survival. Decrease in Bax/Bcl-2 ratio as antiapoptotic signal, reduced the release of cytochrome c from mitochondria into cytoplasm, which further lowered the cleavage of pro-caspase 3 to its activated form caspase-3 (Park et al., 2015). EGCG pretreatment suppressed apoptosis and enhanced the Bcl-2/Bax ratio, which is consistent with its protective role for mitochondria (Bing et al., 2013), and the result is similar to ours. These results showed that PPs inhibited mitochondria-dependent apoptosis pathways induced by IR in splenocytes.

PPs reduced the ionizing radiation-induced the peripheral blood, the liver and the spleen damages in many aspects in ICR mice. However, PPs alone preatment did not produce significant side effects compared to the control group. The main active ingredients of PPs are catechin-3-O-glucose, catechin, epicatechin, massonianoside B, cedrusin, catechin-3-O-rutinoside, massonianoside C, etc. Packer, Rimach & Virgili (1999) reported that Pycnogenol displays greater biologic effects as a mixture than its purified components do individually, indicating that the components interact synergistically. It is previously reported that catechin and epicatechin ameliorate ionizing radiation-induced oxidative stress and injury in vivo and in vitro (Hu et al., 2011; Sinha et al., 2012).

Polyphenols has been involved in a reduced risk of a number of chronic diseases, including cancer, cardiovascular disease and neurodegenerative disorders (Vauzour et al., 2010). Humans consumed 500–1,500 mg of polyphenols per day from foods and beverages, and any acute or lethal toxicity was not observed (Vogiatzoglou et al., 2014). Concord grape juice polyphenols reduce cardiovascular risk factors in a dose-dependent manner with 0–1,500 mg/day (Blumberg et al., 2015). Therefore, it is safe and reasonable for humans to consume 1,500 mg of PPs daily by oral intake.

In conclusion, PPs played a central role in protecting the ICR mice peripheral blood, liver and spleen tissues from radiation hazards by inhibiting the damage of the hematopoietic and immune systems and the antioxidant defense system, and attenuating the redox imbalance and inhibiting mitochondria-dependent apoptosis pathways. Therefore, the current study has revealed that PPs are promising radioprotective reagents.

Supplemental Information

Table S1 antioxidant activities of PPs from Pinus koraiensis

EC50 or RP0.5 of Antioxidant activities of PPs from Pinus koraiensis.

Click here for additional data file.

Data S1 Raw data-Pine polyphenols restrains

Click here for additional data file.

We would like to thank Dr. Yan Diao, Haina Bai, Keli Yun and Junqiang Qiu for the help with animal experiment material collection.

Abbreviations

PPs Pine Polyphenols

SOD Superoxide Dismutase

GSH Reduced Glutathione

CAT Catalase

MDA Malondialdehyde

GSH-Px Glutathione Peroxidases

BSA Bovine Serum Albumin

FBS Fetal Bovine Serum

MTT 3-(4,5-dimethyl-2-thiazolyl)-2,5-diphenyl-2-H-tetrazolium bromide

ROS Reactive Oxygen Species

IR Ionizing Radiations

RBC Red Blood Cell

WBC White Blood Cell

LPS Lipopolysaccharide

ICR Institute of Cancer Research

Additional Information and Declarations

Competing Interests

Author Contributions

Animal Ethics

Data Availability

The authors declare there are no competing interests.

Hui Li conceived and designed the experiments, performed the experiments, analyzed the data, contributed reagents/materials/analysis tools, wrote the paper, prepared figures and/or tables, reviewed drafts of the paper.

Zhenyu Wang conceived and designed the experiments, analyzed the data, contributed reagents/materials/analysis tools, reviewed drafts of the paper.

Yier Xu analyzed the data, contributed reagents/materials/analysis tools, reviewed drafts of the paper.

Guicai Sun reviewed drafts of the paper.

The following information was supplied relating to ethical approvals (i.e., approving body and any reference numbers):

The experimental protocols were approved by Heilongjiang University of Chinese Medicine (SCXK Hei 2008004). All efforts were made to minimize animal suffering.

The following information was supplied regarding data availability:

Data is uploaded as Supplemental Information.

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
