# Peer review of "Pine polyphenols from Pinus koraiensis prevent injuries induced by gamma radiation in mice"

_PeerJ, doi:10.7717/peerj.1870_

## Round 0.1 · original submission · Major Revisions

Dear Authors

Please revise your manuscript in accordance with the reviewers' comments and make a point-to point response.

Reviewer 1 ·

Basic reporting

"No Comments".

Experimental design

"No Comments".

Validity of the findings

"No Comments".

Additional comments

This paper showed in vivo radioprotection of Pine polyphenols from Pinus koraiensis. Generally in vivo data are most important for evaluating radioprotectors. This article contains useful information for radioprotectors. Therefore, this paper is meaningful in this sence. However, this paper does not have enough data. The experiment was done only once at one administration dose and one irradiation dose. For in vivo experiments, we often experience large scattering of the data. Therefore, at least for Pine polyphenols reproducibility should be checked. In addition, dose dependency (both for administration and irradiation) should be examined to have insight into the reaction mechanism. Dose-reducing factor for Pine polyphenols should be added. The data of the effect of administration timing also provide us important information. In this paper, it is only shown preventive effect of pine polyphenols. Till now, there here been many reports concerning the radioprotective compounds treated before irradiation, especially of polyphenols compounds. Part of hematostimulative and radioprotective action of polyphenols are described in Biomed Pharmacother. 2005;59(10):561-70 as well as “Herbal Radiomodulators: Applications in Medicine, Homeland Defence and Space” (Editors: RK Sharma and Rajesh Arora), CABI Publishing, UK, pp. 175-194. Some of these, the authors should be mentioned in Introduction and Discussion section. As for the study on postirradiation protection, more brief description may be preferred. It would be desirable to add the curve of animal survival (Kaplan-Meier), which would be a good indicator of the expressed antioxidant and immunomodulatory effects of Pine polyphenols.
The other points:
1. Tables and Figures are not sufficiently described; from the description of Tables or Figures should be clear all the data such as: the dose, route and time of injection, the number of animals, the statistics without reading the text. The Tables are necessary but the title of the Tables should stand above the Table and additional explanations of Tables (concentration, treatments, number of animals, statically analyses) should be below.
2. Duplicate or triplicate experiments are needed for every group.
3. Whether Pine polyphenols treatment made by gastric tube? What is the basis for the chosen Pine polyphenols concentrations? How this entry is coincides with the daily intake of these compounds in reality? Please indicate the appropriate dose for humans, where it should except for body surface area involved, and many other parameters such as several parameters of biology, including oxygen utilization, caloric expenditure, basal metabolism, blood volume, circulating plasma proteins, and renal function. Please explain in “Material and Methods” and “Discussion” sections.

4. Please unify the way of writing °C
5. There are many errors in typing.
6. Tags for statistical significance should be written as a superscript.
7. Second, minor problems, there are some mistakes about editorial handling, e.g. % must be written close the number as well as ○C. Please unify the way of writing °C. Please use SI unit (200 µL) and separate the SI unit from the number. In vitro and in vivo should be written in italic...... Abbreviations must be explained the first time they are used, both in the Abstract and again in the main text. There are many typographical and grammatical errors. Recheck them carefully!
Please delete text
Table 1(on next page)
Table in WBC, RBC and platelets counts, spleen index and spleen lymphocyte
transformation in mice
Table 1. The effect of PPs pretreatment on radiation-induced the decline in WBC, RBC and
platelets counts in mice. Table 2. The effect of PPs pretreatment on radiation-induced the
damage in spleen index and spleen lymphocyte transformation in mice.

·

Basic reporting

Good

Experimental design

Good

Validity of the findings

Good

Additional comments

The article is a nice presentation. The dose, duration and distance (from source to subject ) of gamma radiation should be mentioned.

·

Basic reporting

see Validity of the Findings!

Experimental design

see Validity of the Findings!

Validity of the findings

English is correct and it is ease to read the extensive text of the manuscript.
The Introduction section is well written, detailed, and suported by relevant literature.
The Material and Methods section is comprenhensive. In line 77 the word America should be replaced by USA.
Experiments in the Results section are correctly organised and presented. Tables and Figures are complete and clear.
The Discussion section confronts the obtained results with relevant literature in an adequate but rather extensive way confirming radioprotection of PPs.
The references are mostly contemporary and properly selected including a few recent publications of the authors.
General comment: The manuscript may well contribute to the knowledge in the field of radioprotection involving PPs as a new finding. The whole text is rather extensive. If this is acceptable to the policy (scope) of the Journal
the manuscript could be accepted for publication.

Additional comments

as written in Valididy of the Findings

---

## Round 0.2 · accepted · Accept

The revised manuscript is acceptable for publication.

Reviewer 1 ·

Basic reporting

-No Comments

Experimental design

-No Comments

Validity of the findings

No Comments

Additional comments

Thanks for the correction. I have no further comments. The manuscript deserves to be published.

·

Basic reporting

good!

Experimental design

good!

Validity of the findings

good!

Additional comments

The revised manuscript is accetable for publication.